# Neurotoxicity of different amyloid beta subspecies in mice and their interaction with isoflurane anaesthesia

**Laura Borgstedt**[1], **Manfred Blobner**[1,2], **Maximilian Musiol**[1], **Sebastian Bratke**[1], **Finn Syryca**[1], **Gerhard Rammes**[1], **Bettina Jungwirth**[2], **Sebastian Schmid**[1,2]*

**1** Department of Anaesthesiology and Intensive Care Medicine, Klinikum rechts der Isar, Technical University Munich, Munich, Germany, **2** Department of Anaesthesiology and Intensive Care Medicine, University Hospital Ulm, Ulm University, Ulm, Germany

* seb.schmid@tum.de

**Data Availability Statement:** All files with the original date are available from the mediaTUM database: https://mediatum.ub.tum.de/1579195.

## Abstract

### Background

The aim of this study was to assess different amyloid beta subspecies' effects on behaviour and cognition in mice and their interaction with isoflurane anaesthesia.

### Methods

After governmental approval, cannulas were implanted in the lateral cerebral ventricle. After 14 days the mice were randomly intracerebroventricularly injected with Aβ 1–40 (Aβ40), Aβ 1–42 (Aβ42), 3NTyr10-Aβ (Aβ nitro), AβpE3-42 (Aβ pyro), or phosphate buffered saline. Four days after the injection, 30 mice (6 animals per subgroup) underwent general anaesthesia with isoflurane. A "sham" anaesthetic procedure was performed in another 30 mice (6 animals per subgroup, 10 subgroups in total). During the next eight consecutive days a blinded assessor evaluated behavioural and cognitive performance using the modified hole-board test. Following the testing we investigated 2 brains per subgroup for insoluble amyloid deposits using methoxy staining. We used western blotting in 4 brains per subgroup for analysis of tumour-necrosis factor alpha, caspase 3, glutamate receptors NR2B, and mGlu5. Data were analysed using general linear modelling and analysis of variance.

### Results

Aβ pyro improved overall cognitive performance (p = 0.038). This cognitive improvement was reversed by isoflurane anaesthesia (p = 0.007), presumably mediated by decreased exploratory behaviour (p = 0.022 and p = 0.037). Injection of Aβ42 was associated with increased anxiety (p = 0.079). Explorative analysis on a limited number of brains did not reveal insoluble amyloid deposits or differences in the expression of tumour-necrosis factor alpha, NR2B, mGlu5, or caspase 3.

**Funding:** The authors received no specific funding for this work.

**Competing interests:** The authors have declared that no competing interests exist.

## Conclusions

Testing cognitive performance after intracerebroventricular injection of different amyloid beta subspecies revealed that Aβ pyro might be less harmful, which was reversed by isoflurane anaesthesia. There is minor evidence for Aβ42-mediated neurotoxicity. Preliminary molecular analysis of biomarkers did not clarify pathophysiological mechanisms.

## 1. Introduction

The accumulation of amyloid beta (Aβ) in the brain is one of the key factors in the pathophysiology of Alzheimer's disease (AD) [1]. Aβ is generated via processing of amyloid precursor protein (APP) in the amyloidogenic pathway. APP can be cleaved in various positions and several post-translational modifications have been identified resulting in different subspecies [2]. In the brains of people living with AD Aβ 1–40 (Aβ40) is most prevalent. However, the ability to accumulate and form oligomers is elevated in Aβ 1–42 (Aβ42) resulting in increased neurotoxicity [3]. Therefore, these two subspecies of Aβ have been investigated thoroughly [4–10]. Other modifications of Aβ by nitration or pyroglutamylation have been described [11, 12]. Aβ in AD patients contains 10–15% of pyroglutamated amyloid beta 1–42 (AβpE3-42, Abeta pyro) and it represents a dominant fraction of Aβ peptides in senile plaques of AD brains [13]. Abeta nitro (3NTyr10-Aβ) is a nitrotyrosinated (or nitrated) form of amyloid beta 1–42 [12] and is found in the cores of amyloid plaques in AD brains [14]. Their impact on neurobehavioural outcome parameters needs further evaluation [13, 14]. As mice and humans show similarities concerning behaviour, memory and learning [15–18] we chose this species in order to ultimately find the best anaesthetic regimen for people living with AD.

An increasing number of aged patients requires surgery and anaesthesia [19–22]. Consequently, more people living with AD undergo general anaesthesia [23]. It is controversially discussed whether or not anaesthesia can trigger or worsen AD in aged patients [24]. However, an interaction between anaesthetics and Aβ has been shown in various studies [4, 25–30]. Some studies suggest a possible link between anesthesia and AD in humans [31, 32], while more recent ones do not [33, 34]. Also, as stated by Lee et al. it is nearly impossible to discriminate the influence of general anesthesia from the effect of surgery itself on the development of AD [35].

The aim of this investigation was to further elucidate the effects of different Aβ subspecies on cognition in mice and their interaction with anaesthetics. Since there is no mouse model displaying pathology derived from post-translationally modified Aβ proteins, we decided to use the method of intracerebroventricular injection (ICV). Previous experiments have demonstrated that intracerebroventricular injection of Aβ oligomers leads to cognitive deficits [36], although this animal model is restricted to amyloidopathy. We assessed cognitive and behavioural function after ICV injection using the modified hole-board test (mHBT).

To further investigate the interaction between anaesthetics and different Aβ subspecies, mice were exposed to isoflurane anaesthesia with a minimal alveolar concentration (MAC) of 1.0. Isoflurane is one of the most extensively studied anaesthetic agents in animal research. It has been shown to induce caspase activation and increase levels of beta-site APP-cleaving enzyme (BACE) *in vivo* in C57/BL6 mice [27]. Furthermore, isoflurane leads to increased oligomerization of amyloid beta *in vitro* and therefore might interact with the different Aβ subspecies.

Since accumulation of amyloid beta leads to neuroinflammation, apoptosis and disruption of the glutamatergic system, we analysed the brain tissue for TNF alpha, caspase 3, NR2B, and

mGlu5 as a secondary objective [37]. We looked for potential molecular mechanisms mediating cognitive and behavioural impairment and the interaction between isoflurane and Aβ.

## 2. Methods

This study was carried out in strict accordance with the recommendations of the Federation of European Laboratory Animal Science Associations (FELASA). The following experimental procedures on animals were approved by the Governmental Animal Care Committee (Regierung von Oberbayern, Maximilianstr. 39, 80538 Munich, Germany, Chair: Dr. B. Wirrer, Registration number: 55.2-1-54-2532-111-12, November 27th, 2012). All surgical procedures were performed under isoflurane anaesthesia and all efforts were made to minimize suffering. Animal welfare was assessed daily.

### 2.1 Surgical procedure: Implantation of intracerebroventricular cannula

60 male 10-week-old C57BL/6N mice (median weight 26.7 g) obtained from Charles River Laboratories (Sulzfeld, Germany) were housed under standard laboratory conditions (specific pathogen free environment, 12 h light/12 h dark cycle, 22°C room temperature, 60% humidity and free access to water and standard mouse chow) 14 days prior to the experiments for acclimatisation.

For induction of general anaesthesia mice were placed in an acrylic glass chamber that had been pre-flushed with 4.0 Vol% isoflurane and 50% of oxygen. After loss of postural reflexes mice were placed on a warming pad (rectal temperature was measured and maintained at 37.5°C) and the stereotactic frame was mounted. General anaesthesia was maintained with 1.6 Vol% Isoflurane (MAC 1.0) and a fraction of inspired oxygen of 50% (FiO2 0.5) administered via a nose chamber. Mice breathed spontaneously during surgery. The skin was shaved, disinfected and after local anaesthesia with 0.5 ml xylocaine 2% a midline incision was performed to expose the bone. Using a computer controlled motorized stereotactic instrument (TSE Systems, Bad Homburg vor der Hoehe, Germany) the insertion point of the cannula (1 mm lateral and 0.3 mm caudal of Bregma) was determined and a small hole (0.8 mm) was drilled. The cannula was placed with a depth of 3 mm using the stereotactic instrument. For further stabilisation a small screw was placed in the scalp and the cannula was cemented to the scalp and the screw. Wound closure was achieved using single stitches and 0.05 mg/kg of buprenorphine were injected intraperitoneally for pain treatment. The mice were then placed in the acrylic glass chamber with 50% oxygen, now without isoflurane, and were monitored until full recovery from anaesthesia. Afterwards the mice were placed in single cages.

### 2.2 Randomization and blinding

After successful implantation of the intracerebroventricular cannula the mice were randomly assigned to one of ten groups (n = 6 mice per experimental group) regarding Aβ subspecies or PBS and isoflurane anaesthesia or sham procedure using a computer-generated randomization list. The experimental groups were designed as follows: Aβ40/sham (n = 6 mice), Aβ40/isoflurane (n = 6 mice), Aβ42/sham (n = 6 mice), Aβ42/isoflurane (n = 6 mice), Aβ nitro/sham (n = 6 mice), Aβ nitro/isoflurane (n = 6 mice), Aβ pyro/sham (n = 6 mice), Aβ pyro/isoflurane (n = 6 mice), PBS/sham (n = 6 mice), PBS/isoflurane (n = 6 mice). The outcome assessor conducting the mHBT and the personnel performing the analysis of biomarkers were blinded to the group assignment.

## 2.3 Injection of amyloid beta

On day 14 after implantation of the cannula the mice were injected with 5 μl of either Aβ42, Aβ40, 3NTyr10-Aβ (Aβ nitro), AβpE3-42 (Aβ pyro), or PBS through the cannula. For this procedure a Hamilton® syringe connected to a plastic tube and a smaller cannula that was inserted into the ICV cannula were used.

Aβ42 (American Peptide Sunnyvale, CA, USA) was suspended in 100% HFIP (Sigma Aldrich, St. Louis, Missouri, United States) to 1 mg/ml and shaken at 37°C for 1.5 h. This solution was aliquoted to 5–50 μg portions and then HFIP was removed by evaporation for 30 minutes using a vacuum concentrator (Thermo Scientific Savant SpeedVac, Thermo Fisher Scientific, Waltham, Massachusetts, United States of America). When completely dry, the peptide aliquots were stored at -20°C. Before injection aliquoted monomeric Aβ42 was warmed in a water bath at 37°C for 10 minutes, then sonicated for 30 s, dissolved in NaOH (20 mmol/l, pH 12.2) and diluted in PBS (1:100) to start the oligomerization process and sonicated for another 30 s, mixed for 30 s, sonicated for 30 s and mixed again for 30 s before being placed on ice. The Aβ solution was used between 15 and 45 minutes after its preparation. It was brought to room temperature before use by loading it into the cannula 10 minutes before administration. Aβ42 concentration of the injected solution (5.0 μl) was 700 nmol/l resulting in a concentration of 100 nmol/l in the cerebrospinal fluid of the mouse. Aβ40 (American Peptide Sunnyvale, CA, USA), Aβ nitro (provided by Clinical Neuroscience Unit, Department of Neurology, University of Bonn, Sigmund-Freud-Strasse 25, 53127 Bonn, Germany), and Aβ pyro (Bachem AG Bubendorf, Switzerland) were dissolved in PBS to reach concentrations of 3200, 700, and 11900 nmol/l in the 5 μl boli that were used for injection, respectively. This resulted in concentrations of 450, 100, and 1700 nmol/l of the corresponding substance in the cerebrospinal fluid of the mouse. The concentrations were chosen in order to reach equipotential concentrations in the cerebrospinal fluid of these four Aβ substances according to their effect on long term potentiation, excitatory postsynaptic potential, and spine density *in vitro* derived from other experiments [38].

## 2.4 Isoflurane anaesthesia

On day 4 after injection of the different Aβ-substances a 2-hour isoflurane anaesthesia was performed in 30 mice. The other 30 mice underwent a sham procedure (total n = 60 mice). After induction as described in 2.1 the mouse was placed with its nose in a nose-chamber and breathed spontaneously with a PEEP of 5 mbar. Temperature was monitored using a rectal probe and maintained at 37.5°C using a heating mat. A subcutaneous electrocardiogram was placed. Heart rate and impedance respiratory rate were monitored. Isoflurane concentration was maintained at 1.6 Vol% corresponding to a MAC of 1.0. Every 15 minutes the depth of anaesthesia was verified with a tail clamp that was kept in place for 1 minute [39]. The "sham" anaesthetic procedure included handling of the animals and placement in the induction box for 10 minutes without exposure to isoflurane.

## 2.5 Cognitive and behavioural testing

Starting on day 5 after the injection mice were tested for cognitive function, behaviour, and social interaction using the mHBT. This test is a combination of a classical hole-board with an open field test, according to an established protocol [40–42]. With this test, a total of 16 different parameters can be observed simultaneously. For the mHBT the hole-board is placed in the middle of the test arena. Ten cylinders are staggered in two lines on the board. Each cylinder contains a small piece of almond fixed underneath a grid that cannot be removed by the animals (S1 Fig). In addition, each cylinder is flavoured with vanilla to attract the animals' attention. Three of the 10 cylinders are baited with a second–approachable–piece of almond and

marked with white tape. The sequence of marked holes is changed according to a protocol every day. We performed testing for 8 consecutive days from day 5 until day 12 after the intra-cerebroventricular injection. Each mouse underwent four trials per day (300 s/trial).

We evaluated two different parameters regarding cognitive performance: Firstly, if mice visited non-baited holes or did not visit baited holes it was summed up as wrong choice total. A higher total number of errors can be interpreted as an impaired reference memory. Secondly, the total time needed to finish the task (time trial) was recorded as a marker for the overall cognitive performance. An extended duration represents a pathological finding. If an animal did not finish the task, i.e. did not find all three pieces of almond within 300 s, the trial was abrupted. Two parameters focussed on anxiety: the latency with which the mice first visited the area of interest (grey board with cylinders, S1 Fig) and the time spent on this board. An increased latency and reduced time on board represent avoidance behaviour and, therefore a higher level of anxiety. Arousal was evaluated by the total time mice spent grooming during the trial. The number of line crossings in the test arena served as an indicator for locomotor activity. The number of visits to a baited hole was counted as correct hole visits, with higher values being associated with increased direct exploratory motivation.

## 2.6 Sampling of brain and blood

On day 13 after ICV injection mice were deeply anaesthetized and brains were harvested by decapitation. The samples were stored at -80°C. Each brain was separated into hemicortices. One hemicortex was sliced into sagittal slides of 50 μm. The other one was separated into pre-frontal motor cortex, sensory cortex and hippocampus.

## 2.7 Amyloid deposits

To detect amyloid deposits, a total of 20 (2 brains of each subgroup) brains were investigated. 50 μm thick sagittal brain slices (n = 21 per brain) including sensory cortex and hippocampus were fixed on microscope slides in −20°C acetone for 20 min. The staining protocol has been described previously [43–45]. After drying at room temperature, each slice was washed twice with wash solution (PBS/Ethanol denaturised with MEK in 1:1 ratio) for 10 minutes. Then meth-oxy-X04 solution (10 mg methoxy-X04 powder (Tocris, Bioscience) diluted in 100 μl Dimethyl-sulfoxid, mixed with 450 μl of 1,2-Propandiol, 450 μl of PBS, and 50 μl 1 N NaOH; 800 μl of this stock was diluted with 200 ml of a 1:1-PBS/ethanol solution) was applied to the slices on a shaker in the dark for 30 minutes. To remove the unbonded methoxy-X04, brain slices were washed three times with wash solution and twice with distilled water for 10 minutes per step. In a final step, brain slices were preserved in fluorescence mounting medium (DAKO, Santa Clara, California, USA). Methoxy-X04 has a high binding affinity for amyloid deposits. The stained brain slices were imaged by magnification using fluorescence microscopy in tile scan mode (ZEISS Axio Imager, ApoTome.2 and Zen 2012 Blue Software, Oberkochen, Germany).

## 2.8 Analysis of tumor necrosis factor (TNF) alpha, caspase 3, N-methyl-D-aspartate-receptor subunit 2B (NR2B), and metabotropic glutamate receptor 5 (mGlu5)

Sensory cortex and hippocampus of four animals per group (total n = 40) were suspended sepa-rately in grinding tubes (Sample Grinding Kit, GE Healthcare, Munich, Germany) and extraction-solution was added (1ml: 970μl Ripa Buffer; 20μl 50xComplete; 10μl 100xPhenylmethylsulfonyl-fluorid; 1μl Pepstation). After centrifugation the supernatant was stored at -80°C. The protein-con-centration (by Bradford Assay) was standardized with Laemmli buffer (1.4ml, 4x times: 1ml

NuPage LDS Sample Buffer (Invitro-gen NP0007); 400μl NuPage Sample Reducing Agent (Invi-trogen NP0009)). The samples were transferred onto the gel (TGX Stain-Free™ FastCast™ Acrylam-ide Kit 10%; Bio-Rad Laboratories GmbH, Munich, Germany) in equal amounts (20μl) and equal protein-concentrations (1μg/μl) per lane for separation by gel electrophoresis and blotted onto a membrane (Amersham Hybond Low Fluorescence 0.2 μm Polyvinylidenfluorid-Membrane; TH Geyer, GmbH, Munich, Germany). The membrane was blocked for one hour and incubated after-wards with the first antibody ("TNF-alpha" ProSci XP-5284 1:1000, "Caspase 3" Cell Signaling #9662 1:1000, "NR2B" Cell Signaling #4207 1:1000, or "mGlu5" Abcam ab53090 1:1000) overnight at 4˚C. After washing it with TBS/T (1l: 1l dH2O; 3g Tris, 11.1g NaCl; 1ml Tween 20) the mem-brane was incubated with the second antibody ("Anti-rabbit IgG" Cell Signaling #7076 1:10 000) for one hour. Following another washing step, the membrane was incubated in 1ml Clarity™ West-ern ECL Substrate (Bio-Rad Laboratories GmbH, Munich, Germany) for 1 minute. The labelled proteins were detected with camera imaging (Bio-Rad Molecular Imager® ChemiDocTM XRS+; Bio-Rad Laboratories GmbH, Munich, Germany). For analysis and normalisation ImageLab® was used in addition to the Stain-Free® Technology to assess the total protein amount (Bio-Rad Laboratories GmbH, Munich, Germany). A standard lane was included in every blot.

## 2.9 Statistical analysis

Neurocognitive and behavioural parameters were analysed using general linear models (GLM) comparing each substance (Aβ40, Aβ42, Aβ nitro or Aβ pyro) to PBS: We analysed the between-group factors subspecies for injection, anaesthesia (isoflurane or sham) and the within-group factor time and their interaction terms. Effects of time were analysed in a linear fashion due to the strictly monotonic decreasing character of learning curves in these tests. For determination of the effect size we calculated mean differences with 95% confidence interval and partial eta-squared with 90% confidence interval.

Regarding sample-size calculations, variables of the mHBT are considered relevant if two groups differ two times the given standard deviation. Based on a type I error of 0.05, a type II error of 0.20 and two-sided t-tests at the final test level of the hierarchical model 4 animals per group would have been appropriate. Our internal standard, however, suggests a minimal group size of six, which we used in our experiment.

In addition, we performed explorative studies on a limited number of brains on amyloid deposits and different biomarkers in order to detect possible mechanisms of interaction. Since distribution of the protein-concentrations of TNF alpha, caspase 3, NR2B and mGlu5 in the western-blot analysis were positively skewed, the statistical analyses were performed following logarithmic transformation. Western-blot results were analysed using analysis of variance (ANOVA) comparing each substance to PBS with the additional factors anaesthesia and the interaction term. The significance level was set at $p < 0.05$. Calculations were done with SPSS Statistics® (Version 24.0; IBM; New York; United States).

## 3. Results

### 3.1 Cognitive and behavioural testing

In mice injected with Aβ pyro the time required to complete the test, i.e. time trial, was decreased (Aβ pyro compared to PBS: mean difference (95% confidence interval): -41 s (-80 to -2 s); partial eta-squared (90% confidence interval): 0.198 (0.006 to 0.413); p = 0.038; Fig 1A). General anaesthesia with isoflurane led to an increase in time trial in those mice compared to control (Aβ pyro isoflurane vs. Sham: 55 s (17 to 94 s); 0.307 (0.055 to 0.508) p = 0.007; Fig 1A). Reference memory function, represented by the total number of wrong choices, was com-parable between the different Aβ substances and PBS (Fig 1B).

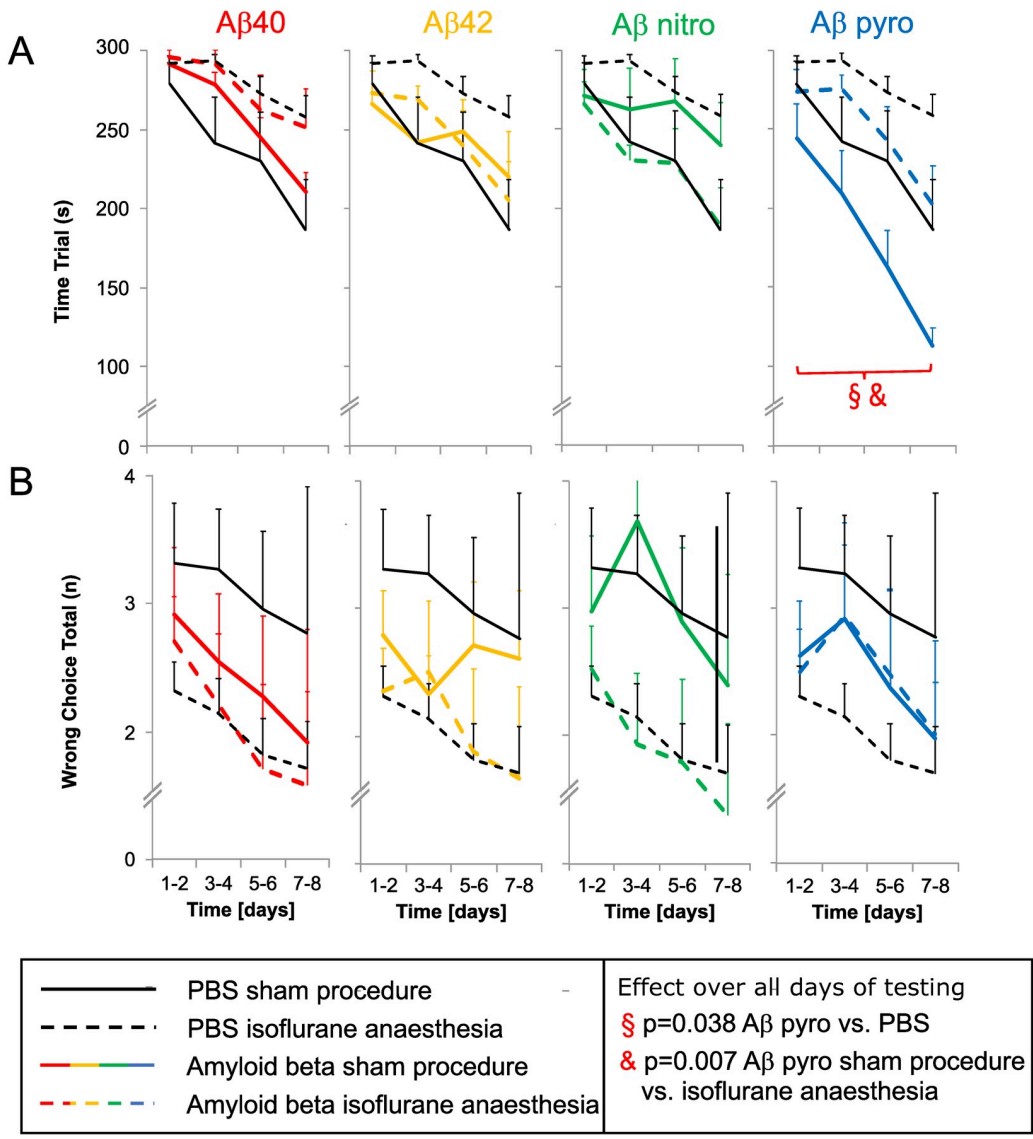

**Fig 1. Neurocognitive function after anaesthesia in mice injected with different Aβ subspecies compared to control.**
A: Time Trial (overall cognitive performance), B: Wrong choices total (declarative memory); mean of all tests on two days and standard error (whiskers).

The time on board the exposed part of the test arena increased over time in mice injected with Aβ40, Aβ nitro, and Aβ pyro (partial eta-squared (90% confidence-interval): 0.340 (0.063 to 0.540), p = 0.007 (Aβ40); 0.226 (0.013 to 0.483), p = 0.029 (Aβ nitro); 0.320 (0.057 to 0.521), p = 0.008 (pyro); Fig 2A), but not in mice injected with Aβ42 (0.180 (0.000 to 0.416), p = 0.079; Fig 2A). Regarding the latency the mice first entered the board, another parameter for anxiety, there was no difference between the different Aβ substances and PBS (Fig 2B). In animals injected with Aβ pyro an isoflurane anaesthesia decreased the time on board (mean difference (95% confidence interval): -12 s (-22 to -2 s); partial eta-squared (90% confidence interval): 0.245 (0.021 to 0.460), p = 0.022; Fig 2A) and increased the latency until the animals first visited the board (33 s (2 to 65 s); 0.200 (0.007 to 0.416), p = 0.037; Fig 2B).

Locomotor activity (line crossings) and direct exploratory motivation (correct hole visits), representing further behavioural parameters, did not differ between groups (Fig 3A and 3B).

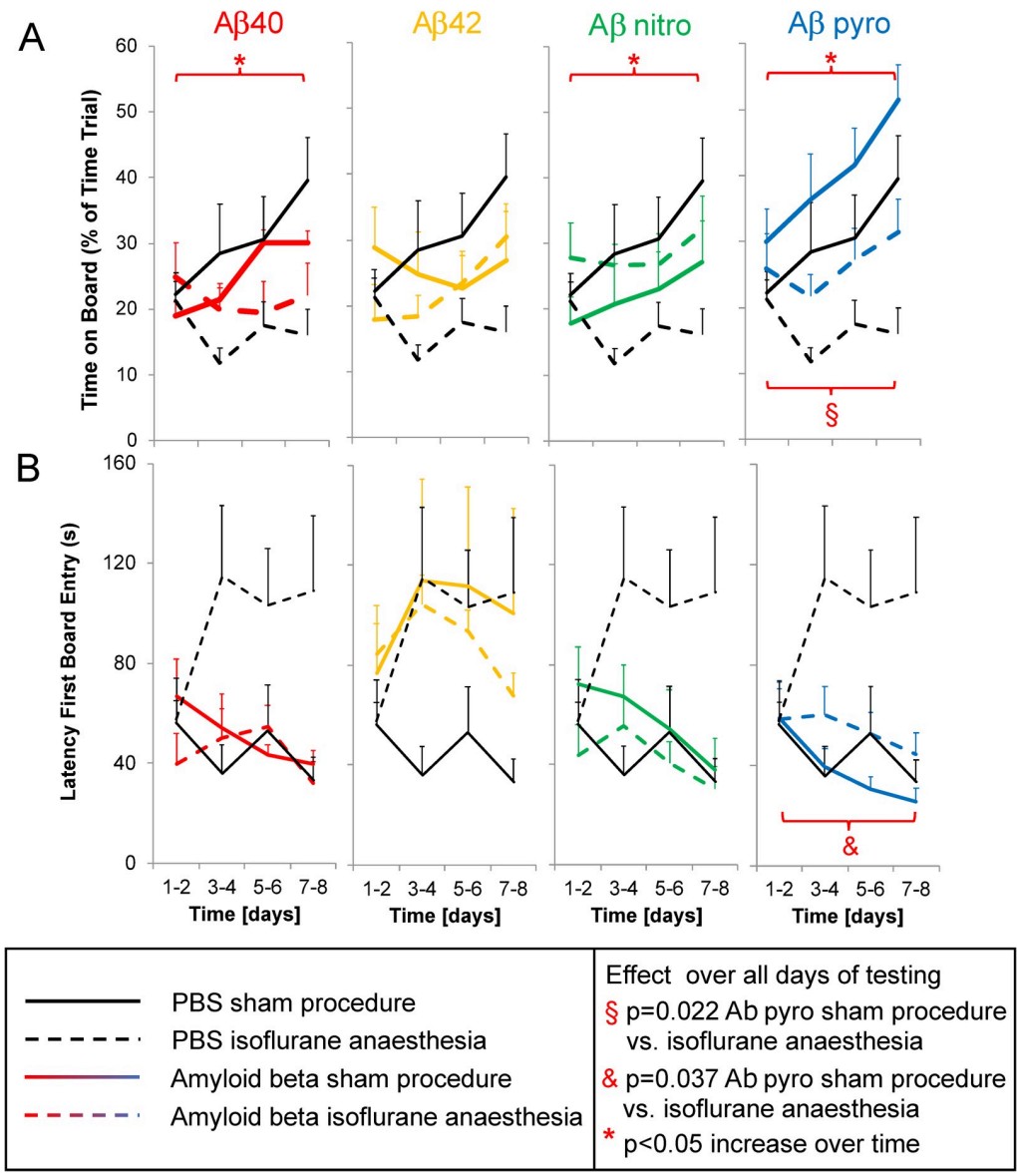

**Fig 2. Anxiety-related behavioural changes after anaesthesia in mice injected with different Aβ subspecies compared to control.** A: Time on Board, B: Latency First Board Entry (both anxiety); mean of all tests on two days and standard error (whiskers).

## 3.2 Amyloid deposits

There were no insoluble amyloid deposits present on day 13 after the intracerebroventricular injection of the different subspecies in both groups, with and without isoflurane exposure.

## 3.3 Analysis of TNF alpha, caspase 3, NR2B, and mGlu5

There was no difference in protein concentrations of TNF alpha, caspase 3, NR2B, and mGlu5 in sensory cortex or hippocampus between the different subspecies compared to PBS. Isoflurane anaesthesia did not have an effect on these protein concentrations (Fig 4A–4D).

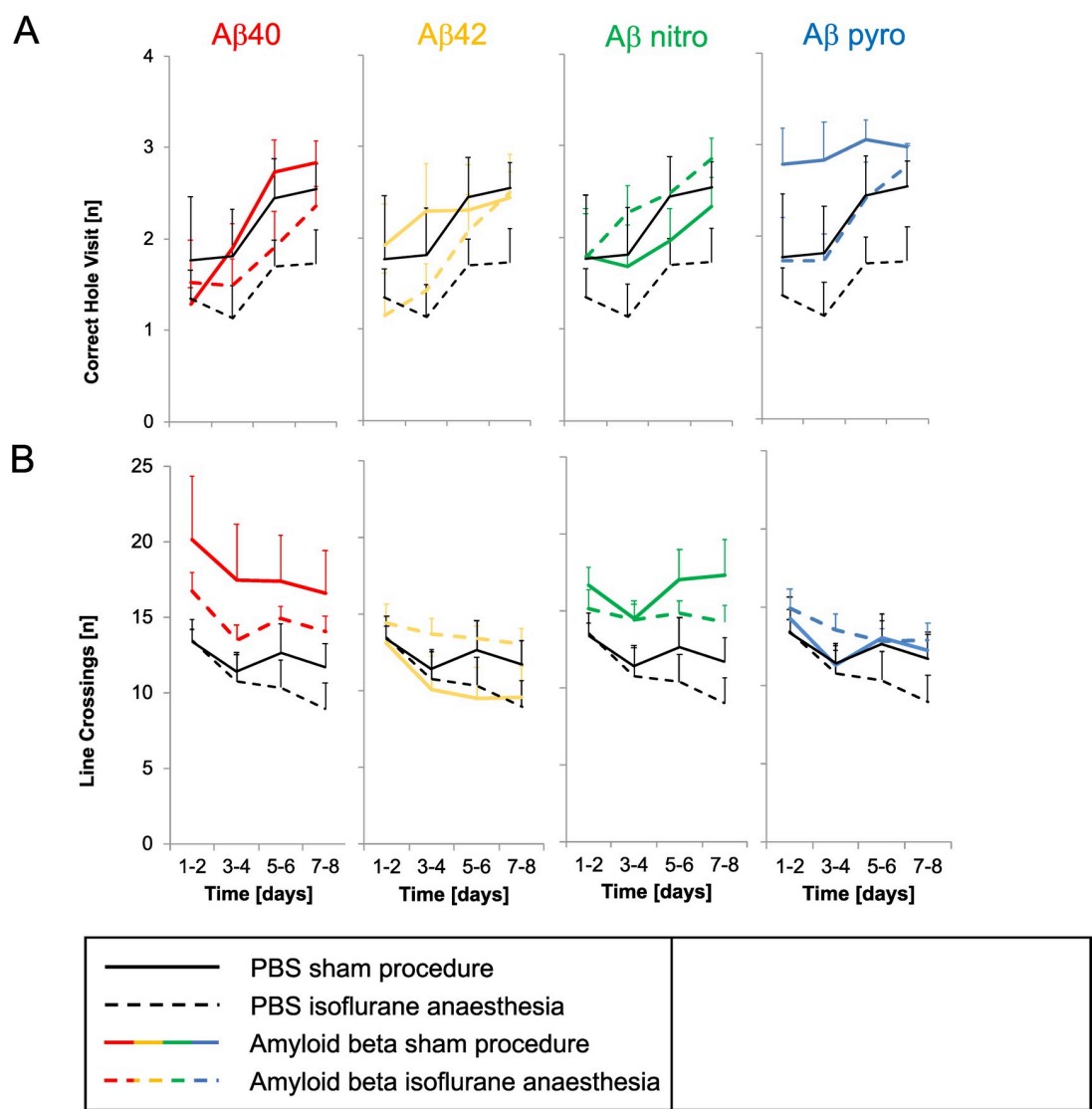

**Fig 3. Behavioural parameters of mice injected with different Aβ subspecies after anaesthesia compared to control.** A: Correct Hole Visits (exploratory motivation), B: Line Crossings (locomotor activity); mean of all tests on two days and standard error (whiskers).

## 4. Discussion

Modified hole-board testing revealed that, shortly after ICV injection, Aβ pyro may be less harmful, as it is associated with an enhancement of overall cognitive performance. This improvement was reversed by isoflurane anaesthesia, as the interaction between isoflurane and Aβ pyro led to decreased exploratory behaviour. The mice spent less time on board and took longer before entering the board or finishing the trial. Aβ42 led to increased anxiety. This might be explained by an elevated toxicity compared to the other Aβ subspecies, since Aβ42 is thought to be more pathogenic as it forms toxic oligomers more readily than other Aβ subspecies [46, 47]. We were not able to detect insoluble amyloid deposits in a limited number of brains. The preliminary analysis of biomarkers for apoptosis, inflammation, and the glutamate receptor subunits NR2B and mGlu5 did not reveal correlations to the different Aβ subspecies or isoflurane anaesthesia.

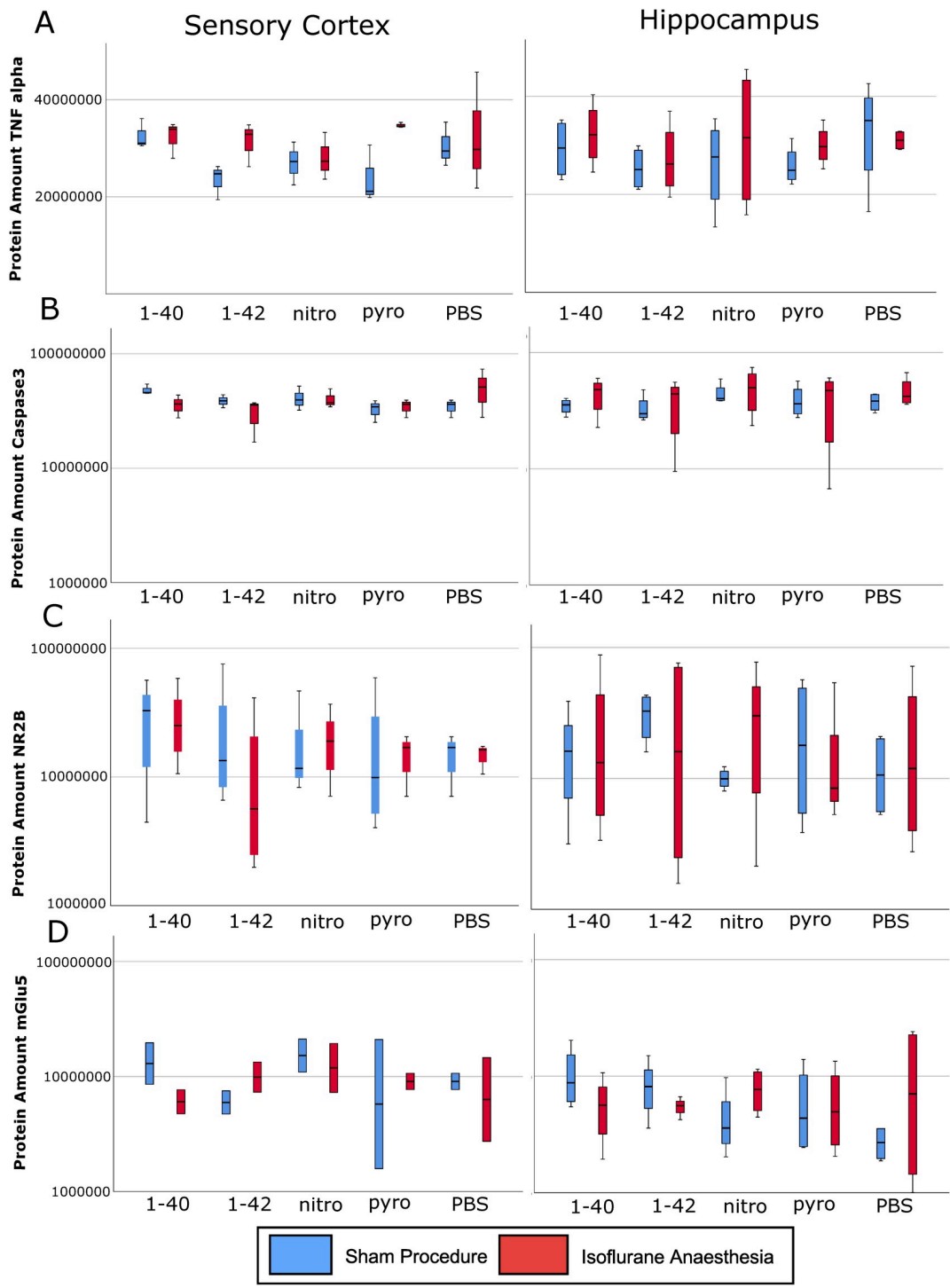

**Fig 4. Total protein amount in sensory cortex and hippocampus.** A: Tumor necrosis factor alpha (TNF alpha), B: Caspase 3, C: N-methyl-D-aspartate receptor subunit 2B (NR2B); D: Metabotropic glutamate receptor 5 (mGlu5); median (horizontal lines), interquartile range (box) and range (whiskers).

To further investigate the pathophysiology of AD and potential therapeutic options, different mouse models have been developed in the last years. Several transgenic mice are available,

showing cognitive and behavioural impairment comparable to the pathological changes in human AD patients [48]. Whereas first animal models focussed on amyloidopathy, more recent transgenic mouse models also address aspects of tau causality [49]. In contrast, the ICV injection as a well-accepted method to simulate AD-like pathology is also restricted to amyloidopathy. Our data show that after a short period of five days after the injection of different Aβ subspecies in the lateral ventricle of mice there are only minor changes to cognitive or behavioural parameters. Our model was not able to display more complex neurological changes of AD like memory loss or learning impairment.

The main objective of our study was to elucidate the neurotoxicity of different Aβ subspecies and their interaction with isoflurane *in vivo*. Presently there is no AD mouse model available which represents an Aβ derived pathology including post-translationally modified Aβ proteins. Therefore, we decided to use the intracerebroventricular injection model being aware of its focus on amyloidopathy. In previous experiments we were able to show a cognitive impairment when testing of the mice began on day 2, 4, or 8 after intracerebroventricular injection [36]. Therefore, we performed isoflurane anaesthesia or a sham procedure on day 4 after the intracerebroventricular injection and started testing the following day. We expected a maximum interaction between anaesthesia and Aβ during this timepoint of maximum cognitive impairment. However, modified hole-board testing only showed minor alterations in cognitive performance. Mice injected with Aβ pyro showed an improved overall cognitive performance probably mediated by a decrease in anxiety over time. This finding of a supposedly reduced harmful effect on cognitive performance of Aβ pyro contrasts with other studies on the one hand. For Aβ pyro an increased potential to aggregate has been shown and therefore, it is considered as an Aβ subspecies with high neurotoxicity [38, 50]. In 2012, Nussbaum et al. published a potential mechanism by which Aβ pyro could trigger AD pathogenesis [51]. Their study showed a higher toxicity of Aβ pyro in wildtype (WT) mice neurons *in vitro* by co-oligomerization with excess Aβ42 as well as neuron loss and gliosis in WT mice at the age of 3 months in a tau-dependent manner. We examined the effect of Aβ pyro in 10-week-old mice without excess Aβ42 and in a tau null background which might explain the improvement in overall cognitive performance. On the other hand, we might have observed a neuroprotective effect of Aβ pyro. Emerging evidence suggests that Aβ might work in a neuroprotective way as an antioxidant, metal chelator, or by increasing synaptic plasticity, preventing excitotoxicity and stimulating learning and memory [52].

Furthermore, in our study we investigated the effect on cognitive performance after a very short period after ICV injection. At this early timepoint the increased aggregation-potential might not have led to clinically relevant oligomers, which are responsible for neurotoxicity [53]. Even on day 13 after injection we were not able to detect insoluble amyloid deposits.

In the mice injected with Aβ pyro, an exposure to isoflurane results in increased behavioural signs of anxiety. This impairs the improved overall cognitive performance and reduces it to baseline. A possible explanation might be an enhanced oligomerisation given the fact that an isoflurane anaesthesia enhances oligomerization and cytotoxicity of Aβ *in vitro* and has a negative effect on cognitive performance and mortality *in vivo* [4, 54]. A potential mechanism of interaction might lie in the ability of aggregated Aβ pyro to form membrane pores and the fact that Aβ pyro and isoflurane are both hydrophobic agents [55–57]. Several other studies report both favourable and non-favourable interactions between isoflurane and especially Aβ40 and Aβ42 [58–60]. We were not able to confirm these findings in our experiments using a non-transgenic mouse model.

Besides Aβ pyro, Aβ42 is also considered as one of the most neurotoxic subspecies with a high potential for aggregation [61]. Negative effects on cognitive function and behavioural parameters have been shown in several investigations [36, 62] but also dose-dependent effects

have been reported. In 2017, Lazarevic et al. showed that 200 pM of Aβ40 and Aβ42 had a stimulating effect on neuronal synapses whereas 1 μM of Aβ40 and Aβ42 had a decreasing effect on active synapses [63]. In our study, increased anxiety in mice injected with Aβ42 was the only measurable effect of this Aβ subspecies. We did not see any effects on cognitive function, which contrasts with our former results, where we saw negative effects even very shortly after ICV injection [36]. This might be explained by an interference of increased anxiety with the cognitive testing in the mHBT [40]. A limitation regarding these findings is the fact, that in contrast to the other Aβ subspecies Aβ42 was dissolved in NaOH. Although the concentration was very low, NaOH might have had an additional inhibiting effect on the parameters of the mHBT.

We also did explorative studies on a limited number of brains on amyloidopathy, in order to detect targets for future research concerning the pathomechanism of different Abeta subspecies and the interaction of isoflurane and amyloid beta. As we were not able to detect insoluble amyloid deposits, again, the short time period between injection and testing might not be sufficient for Aβ42 to oligomerize.

As inflammation and apoptosis are considered as main driving forces behind the pathology of AD we examined the brains for changes in TNF alpha and caspase 3 as representative biomarkers [37]. Besides, alterations in the glutamatergic system like deficiencies in synaptic NR2B subunit phosphorylation and an accumulation and over-stabilization of mGlu5 are also considered as factors promoting the development of AD and might have been changed in our animals [64–66]. Since the intracerebroventricular injection only had minor effects on cognitive and behavioural parameters, not surprisingly, we did not detect significant changes in these biomarkers. These data should be considered preliminary as the primary endpoint of this study was the cognitive and behavioural outcome.

It might be considered a limitation of our investigation that we analysed biomarkers and amyloid deposits at one time-point after and not consecutively during the modified hole-board testing. However, we did not want to take series of blood samples or perform other analysis procedures which could have influenced neurocognitive testing. We performed the injection in 10-week-old mice, which might be another restriction to our experiments. There might be different interactions between the brain and the Aβ subspecies in older animals, as older brains show a higher amount of free metals as well as a reduction in antioxidative defence [67]. We limited our investigation to the four mentioned subspecies, although several other post-translational modifications of Aβ have been identified. In addition, the interaction of other anaesthetics like desflurane, which was associated with less impact on Aβ in human cerebrospinal fluid, could be investigated in future research [68].

## 5. Conclusions

In conclusion, the model of intracerebroventricular injection is not suitable to simulate the complex symptoms of AD. Analysis of different Aβ subspecies revealed that shortly after ICV injection Aβ pyro might be less harmful, which was reversed by an exposure to isoflurane. There is minor evidence for an increased toxicity of Aβ42. Analysis of biomarkers in a limited number of animals did not clarify pathophysiological mechanisms.

## Supporting information

**S1 Fig. Modified hole-board consisting of test arena and hole-board with cylinders.**
(PDF)

## Acknowledgments

We are indebted to Andreas Blaschke for performing parts of the analysis procedure.

## Author Contributions

**Conceptualization:** Gerhard Rammes, Bettina Jungwirth, Sebastian Schmid.

**Data curation:** Sebastian Schmid.

**Formal analysis:** Laura Borgstedt, Manfred Blobner, Maximilian Musiol, Sebastian Bratke, Finn Syryca, Bettina Jungwirth, Sebastian Schmid.

**Investigation:** Laura Borgstedt, Maximilian Musiol, Sebastian Bratke, Finn Syryca, Sebastian Schmid.

**Methodology:** Manfred Blobner, Bettina Jungwirth, Sebastian Schmid.

**Resources:** Manfred Blobner.

**Supervision:** Manfred Blobner, Bettina Jungwirth.

**Validation:** Bettina Jungwirth, Sebastian Schmid.

**Writing – original draft:** Laura Borgstedt, Bettina Jungwirth, Sebastian Schmid.

**Writing – review & editing:** Laura Borgstedt, Manfred Blobner, Maximilian Musiol, Sebastian Bratke, Finn Syryca, Gerhard Rammes, Bettina Jungwirth, Sebastian Schmid.

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
