## [Decision Letter · Decision Letter 0]

11 Jun 2020

PONE-D-20-10009

Neurotoxicity of different amyloid beta subspecies in mice and their interaction with isoflurane anaesthesia

PLOS ONE

Dear Dr. Schmid,

Thank you for submitting your manuscript to PLOS ONE. After careful consideration by 2 Reviewers and an Academic Editor, all of the critiques of both Reviewers must be addressed in detail in a revision to determine publication status. If you are prepared to undertake the work required, I would be pleased to reconsider my decision, but revision of the original submission without directly addressing the critiques of the two Reviewers does not guarantee acceptance for publication in PLOS ONE. If the authors do not feel that the queries can be addressed, please consider submitting to another publication medium. A revised submission will be sent out for re-review. The authors are urged to have the manuscript given a hard copyedit for syntax and grammar.

**Comments to the Author**

1. Is the manuscript technically sound, and do the data support the conclusions?

Reviewer #1: Partly

Reviewer #2: Partly

2. Has the statistical analysis been performed appropriately and rigorously? 

Reviewer #1: Yes

Reviewer #2: Yes

3. Have the authors made all data underlying the findings in their manuscript fully available?

Reviewer #1: Yes

Reviewer #2: Yes

4. Is the manuscript presented in an intelligible fashion and written in standard English?

Reviewer #1: Yes

Reviewer #2: Yes

5. Review Comments to the Author

Reviewer #1: Rationale of the study

The rationale of the study is partly explained, but many aspects of this study are not put in context in the Introduction. The Introduction could be improved by addressing the comments below.

• What is the relevance of Abeta-pyro and Abeta-nitro to AD? Literature should be included in the introduction (and referenced) to place this in context. Also, what species are Abeta-pyro and Abeta-nitro Abeta1-40 or 1-42?

• “to ultimately find the best anaesthetic regimen for … AD”, this statement in the introduction should be clarified as only one anaesthetic agent is included in this study. Furthermore, an explanation of why isoflurane chosen rather than another aesthetic to study, would strengthen the introduction.

• What is the rationale for investigating the interaction between different Abeta species and anaesthetics? Please include relevant literature in the introduction to address this.

• The rationale for why TNFalpha, caspase 3, NR2B and mGlu5 were chosen as biomarkers is explained in the Discussion, but it would benefit the reader if this information was presented earlier in the manuscript.

• The anaesthetics/AD literature is currently not adequately referenced in the Introduction (e.g. Line 50, 54, 56, 57, 269), this should be rectified.

Methods

The major concerns I have with this work is the experimental design; the study is likely underpowered, lacks some controls and uses different concentrations of the Abeta species across the experimental groups.

• This study is underpowered, behavioural data has n=6 per experimental group, amyloid deposit analysis n=2 per experimental group and WB analysis n=4 per experimental group. An n=2 per experimental group is not sufficient for analysis. Power calculations should be performed to determine the change that can be detected with 90% power for each dataset in the manuscript and this information needs to be included in the manuscript. This will allow the authors to comment on whether the study was sufficiently powered to detect a change, or whether the results a likely to represent false negatives.

• The Abeta 1-42 solution injected includes hexafluoroisopropanol and NaOH, whereas all other Abeta species are diluted in PBS. An additional control group with the same concentration of hexafluoroisopropanol and NaOH in PBS should be included in the study.

• Different concentrations of each Abeta species are used, the explanation for this is that the different concentrations of Abeta species will have equivalent “neurotoxicity” and a paper examining LTP, EPSC and spine density data from brain slices is references. Please define “neurotoxicity” in this context. How do the authors know that this measure of “neurotoxicity” from brain slices will be the same in vivo?.

• Insufficient detail in the methods:

o Where were the Abeta species purchased from or how were they made?

o Concentration of hexafluoroisopropanol in the Abeta 1-42 solution should be provided.

o What was the oligomerization status of all 4 Abeta species at the time of injection? Was it comparable?

o Concentrations of antibodies used for WB should be provided.

o Why were the hippocampus and sensory cortex chosen for WB analysis?

o How many saggital slices were analysed per mouse? Which brain regions were analysed?

Results and Interpretation

• The results are currently unclear and could be improved:

o Are the significant differences in the time trial data for the Abeta-pyro condition only for the 7-8th day of testing? Is the statistically significant difference between Abeta-pyro vs PBS for the sham or isoflurane condition or both?

o Are the data described in line 232 and 233 (Fig 2A) of the manuscript for the sham condition, the anaesthetic condition or both? Is the difference only present for the 7-8th day of testing?

• Abeta-pyro is by far the most concentrated Abeta species when injected, could this explain why the other Abeta species had no/limited effects? And explain the interaction of only Abeta-pyro and isoflurane in the behavioural testing data? Opposite effects of picomolar and micromolar Abeta1-42 and Abeta1-40 have previously been reported (Lazarevic et al., 2017, doi: 10.3389/fnmol.2017.00221). There are also many published papers reporting protective effects of Abeta that should be included in the discussion related to the Abeta-pyro results (i.e. Carrillo-Mora et al., 2014, doi: 10.1155/2014/795375).

• Is there a potential mechanism for the interaction between Abeta-pyro and isoflurane to explain the results?

• Abeta1-42 is injected containing hexafluoroisopropanol and NaOH, there is no control group that accounts for the hexafluoroisopropanol and NaOH, thus, the difference in the time spent on the exposed part of the arena in the Abeta1-42 experimental group (Figure 2A) may be due to the impact of hexafluoroisopropanol, NaOH, Abeta1-42 or a combination of the three.

• The lack of difference in the amyloid deposits and TNFalpha, caspase 3, NR2B and mGlu5 may be due to the study being underpowered or a non-optimal concentration of Abeta species being used. This should be discussed.

• What were the differences between the previous study Schmid et al., 2017 (Reference 8) and the current study?

The anaesthetic used for canula insertion surgery is different, the concentration of Abeta 1-42 injected is much higher, the number of mice per experimental group is higher, this warrants discussion.

Other

• The manuscript should use inclusive language for people living with dementia (i.e. “patients suffering AD” is not appropriate, “people living with AD” is appropriate). This should be revised throughout.

• Figure 1 – reduce the range of the y axis so that data and error bars can be seen clearly.

• Figure 1, 2 – marking of significance on graphs is unclear, which time point is this for? Please rectify.

• Line 232 remove the word “as”.

• Line 268 “…an inhibiting effect on behaviour” this should be reworded to be more specific.

• Paragraphs 2 and 3 of the Discussion could be integrated for a better justification of the mouse model used.

• “in vitro” and “in vivo” should be italicised throughout.

Reviewer #2: The present study investigated the effect of different amyloid beta subspecies on behaviour and cognition in mice and their interaction with isoflurane anesthesia. The main findings of the study are that Aβ pyro improved overall cognitive performance which seemed to be contrary to the previous studies, and isoflurane could counteract this improvement. Inflammation and apoptosis biomarkers such as tumor -necrosis factor alpha, NR2B, mGlu5, or caspase 3 were not involved in this process. There are several points which the authors should consider.

1. The biochemical endpoints appear to be limited to a single evaluation This would be informative to have additional time points to better understand the effects on these markers.

2. Isoflurane is rarely used clinically, as it is known for its adverse effects on cognitive functions. It would be more pellucid if the authors could illuminate the reason why selected isoflurane in this study.

3. The size of the experimental groups of mice in the study is unclear. Is the number of animals included based on power analysis? Please state the number of animals in each experimental group in Material and Methods

6. PLOS authors have the option to publish the peer review history of their article (what does this mean?). If published, this will include your full peer review and any attached files.

**Do you want your identity to be public for this peer review?** For information about this choice, including consent withdrawal, please see our Privacy Policy.

Reviewer #1: No

Reviewer #2: No

We look forward to receiving your revised manuscript.

Kind regards,

Stephen D. Ginsberg, Ph.D.

Section Editor

PLOS ONE

---

## [Author Response · Author response to Decision Letter 0]

4 Nov 2020

Response to Reviewers

Reviewer #1: Rationale of the study

The rationale of the study is partly explained, but many aspects of this study are not put in context in the Introduction. The Introduction could be improved by addressing the comments below.

• What is the relevance of Abeta-pyro and Abeta-nitro to AD? Literature should be included in the introduction (and referenced) to place this in context. Also, what species are Abeta-pyro and Abeta-nitro Abeta1-40 or 1-42?

Thank you very much for this important comment. Abeta-pyro, also known as AβpE3-42, has first been described in 1992 by Mori et al. [1]. It is a modification of Abeta 1-42. Total Abeta contains 10-15% of pyroglutamated amyloid beta (i.e. Abeta pyro) and it represents a dominant fraction of Aβ peptides in senile plaques of AD brains [2]. Abeta nitro (3NTyr10-Aβ) is also a nitrotyrosinated (or nitrated) form of amyloid beta 1-42 [3] and is found in the cores of amyloid plaques in AD brains [4]. Pyroglutamylation as well as nitrotyrosination of amyloid beta leads to increased oligomer stability and thus neurotoxicity in vitro as well as in vivo shown by neurodegeneration, premature mortality of mice, and disruption of calcium dyshomeostasis [5, 6]. 

Abeta1-40, Abeta1-42, Abeta-pyro and Abeta-nitro are all post-translationally modified forms of the peptide amyloid-beta. 

We described Abeta pyro and Abeta nitro in lines 54 – 58 (unmarked version of the revised manuscript)/57 – 61 (marked-up copy of the revised manuscript) more clearly and cited relevant literature.

• “to ultimately find the best anaesthetic regimen for … AD”, this statement in the introduction should be clarified as only one anaesthetic agent is included in this study. Furthermore, an explanation of why isoflurane chosen rather than another aesthetic to study, would strengthen the introduction.

We are grateful to Reviewer #1 for this extremely helpful remark. With this sentence we wanted to express our long-term goal of understanding the pathophysiology underlying Alzheimer’s disease and its interaction with different anaesthetics. Since the aim of our study was to investigate the effects of different intracerebroventricularly administered Abeta subspecies (Abeta1-40, Abeta1-42, Abeta-pyro and Abeta-nitro) in vivo and their interaction with general anaesthesia, we chose to concentrate on one anaesthetic agent, i. e. isoflurane. Isoflurane has been shown to induce caspase activation and increase levels of beta-site APP-cleaving enzyme (BACE) in vivo in C57/BL6 mice [7]. Sevoflurane seems to induce cellular and histological effects comparable to isoflurane [8], while desflurane was associated with a decrease in Abeta 1-42 levels [9]. 

As recommended, we included parts of the paragraph above in the introduction (lines 78–82 unmarked version/lines 92–96 marked-up version of the revised manuscript) and discussed desflurane as a possible anesthetic for future research (lines 404-406 unmarked version/lines 429-431 marked-up version of the revised manuscript).

• What is the rationale for investigating the interaction between different Abeta species and anaesthetics? Please include relevant literature in the introduction to address this. 

Thank you for this important critique which helps us to improve our manuscript. Alzheimer’s disease (AD) is the most common form of dementia worldwide and affects as much as three percent of men and women aged between 65 and 74 years [10]. Due to an ongoing medical progress this population is also very likely to undergo surgery, often conducted under general anesthesia [10]. It is still unclear whether general anesthesia contributes to the development of AD. Some studies suggest a possible link between anesthesia and AD [11, 12], while more recent ones do not [13, 14]. Also, it is nearly impossible to discriminate the influence of general anesthesia from the effect of surgery itself on the development of AD, as Lee et al. stated earlier this year [15]. To further illuminate the pathophysiology behind AD and the possible association of anesthesia and AD we wanted to investigate the potency of inducing AD and their individual interaction with isoflurane of the most prominent amyloid beta subspecies (i. e. Abeta 1-40, Abeta 1-42, Abeta pyro and Abeta nitro) one by one. 

We included relevant literature in the introduction in lines 62, 65, 66, 68 (unmarked copy of the revised manuscript)/67, 70 - 73 (marked-up version of the revised manuscript).

• The rationale for why TNFalpha, caspase 3, NR2B and mGlu5 were chosen as biomarkers is explained in the Discussion, but it would benefit the reader if this information was presented earlier in the manuscript.

We included our reasons for investigating TNFalpha, caspase 3, NR2B and mGlu5 in the introduction (lines 83 - 85 (unmarked version)/102 - 104 (marked-up version)). 

• The anaesthetics/AD literature is currently not adequately referenced in the Introduction (e.g. Line 50, 54, 56, 57, 269), this should be rectified.

Thank you for raising this important point. We corrected for this in the unmarked version of the revised manuscript in lines 52, 53, 59, 61, 62, 64, 317/in the marked-up copy of the revised manuscript in lines 55, 56, 63, 66, 67, 70, 340. We also reworded lines 314 - 317 in the unmarked copy of the revised manuscript/338 - 340 in the marked-up version of the revised manuscript. 

Methods

The major concerns I have with this work is the experimental design; the study is likely underpowered, lacks some controls and uses different concentrations of the Abeta species across the experimental groups.

• This study is underpowered, behavioural data has n=6 per experimental group, amyloid deposit analysis n=2 per experimental group and WB analysis n=4 per experimental group. An n=2 per experimental group is not sufficient for analysis. Power calculations should be performed to determine the change that can be detected with 90% power for each dataset in the manuscript and this information needs to be included in the manuscript. This will allow the authors to comment on whether the study was sufficiently powered to detect a change, or whether the results a likely to represent false negatives.

We apologize for not describing the statistical approach in sufficient detail, especially the sample size considerations. Accordingly, the reviewer must have come to the impression of a power problem. We included our sample size considerations in the methods section (lines 245 - 249 in the unmarked version of the revised manuscript/ lines 268 - 272 in the marked-up version of the revised manuscript):

The primary endpoint of the study was the cognitive and behavioural outcome. The variables of the hole-board test are considered relevant if two groups differ two times the given standard deviation. Based on a type I error of 0.05, a type II error of 0.20 and two-sided t-tests at the final test level of the hierarchical model 4 animals per group would have been appropriate. Our internal standard, however, suggests a minimal group size of six, which has been used accordingly.

In order to facilitate readability of our findings in the modified hole-board test, we calculated the effect size of our findings and included mean differences with 95% confidence interval and partial eta-squared in the manuscript (unmarked version of the revised manuscript: lines 242 – 244, 262, 263, 275, 281/ marked-up version of the revised manuscript: lines 266, 267, 285, 286, 298, 304). 

We agree with the reviewer, that our study was not sufficiently powered for amyloid deposit and western blot analysis. We did explorative studies on a limited number of brains on amyloidopathy and other biomarkers in order to detect targets for future research concerning the pathomechanism of different Abeta subspecies and the interaction of isoflurane and amyloid-beta. Since we could not detect any amyloid deposits, we did not perform statistical analyses regarding amyloidopathy. Although we are aware of the shortcomings, we think that the results of the amyloid deposit and the western blot analysis are valid and should be presented in the manuscript. However, we stated the preliminary character of these analyses more clearly in the abstract and the discussion of the revised manuscript (abstract: lines 25, 26, 32, 38 of the unmarked version/lines 25 - 27, 34, 40 of the marked-up copy; discussion: lines 317, 318, 382, 394, 411 of the unmarked version/lines 273, 341, 406, 419, 436, 437 of the marked-up copy).

• The Abeta 1-42 solution injected includes hexafluoroisopropanol and NaOH, whereas all other Abeta species are diluted in PBS. An additional control group with the same concentration of hexafluoroisopropanol and NaOH in PBS should be included in the study.

We are deeply grateful to Reviewer #1 for raising this important point and apologize for providing insufficient detail in the original version of the manuscript. 

After suspending Abeta 1-42 in hexafluoroisopropanol (HFIP), it was removed from the stock solution using a vacuum concentrator (Thermo Scientific Savant SpeedVac, Thermo Fisher Scientific, Waltham, Massachusetts, United States of America). HFIP has been shown to increase cell permeability and to decrease cell viability [16, 17], so a protocol to purge HFIP from the Abeta 1-42 stock solution was established in our lab [18]. We reworded the respective part in the Methods section in the revised manuscript (unmarked version: lines 133 - 138, marked-up version: lines 154 - 159). 

After Abeta 1-42 was dissolved in NaOH it was diluted 1:100 with PBS. Therefore, the amount of NaOH injected can be considered as minimal. We added the dilution ratio in the manuscript (unmarked version: line 140, marked-up version: line 162). As we did not shield the solution containing NaOH from air, NaOH reacted with CO2 in the 15 to 45 minutes before injection and formed sodium carbonate and sodium bicarbonate, which further reduced the amount of NaOH. In conclusion we consider the effect of the NaOH used to solve Abeta 1-42 minimal. To perform our experiments in accordance with the principles of the 3Rs (Replacement, Reduction and Refinement) in animal research we decided against an additional control group. However, in order to account for a possible bias, we added the use of NaOH in only one group as a possible limitation in the discussion of the revised manuscript (unmarked version: lines 378 - 381, marked-up version: lines 402 - 405).

• Different concentrations of each Abeta species are used, the explanation for this is that the different concentrations of Abeta species will have equivalent “neurotoxicity” and a paper examining LTP, EPSC and spine density data from brain slices is references. Please define “neurotoxicity” in this context. How do the authors know that this measure of “neurotoxicity” from brain slices will be the same in vivo?

Thank you very much for drawing our attention to this important point. Our definition of “neurotoxicity” in this context was the ability of Abeta 1-40, Abeta 1-42, Abeta pyro and Abeta nitro to induce alterations in behaviour, cognition and fine motor skills in mice and the ability to induce amyloid deposits in the brains, respectively. With our study we wanted to investigate whether the findings of Rammes et al. (Rammes et al., “The NMDA receptor antagonist Radiprodil reverses the synaptotoxic effects of different amyloid-beta (Aβ) species on long-term potentiation (LTP)” Neuropharmacology. 2018 Sep 15;140:184-192. doi: 10.1016/j.neuropharm.2018.07.021. Epub 2018 Aug 11.) concerning the AD-inducing effects of different Abeta subspecies in vitro could be transferred into a mouse model. Therefore, we chose to use the same concentrations as Rammes et al. We did not know if the effects on brain slices were to be the same in vivo but we wanted to find out with this study. In order to avoid confusion for the reader we replaced the term “neurotoxicity” in line 152 - 153 (unmarked version of the revised manuscript)/line 175, 176 (marked-up copy of the revised manuscript) and replaced it with the actual in vitro findings of the cited reference. 

• Insufficient detail in the methods:

o Where were the Abeta species purchased from or how were they made?

We apologize for not stating this beforehand. Aβ1-40 and Aβ1-42 were both purchased from American Peptide Sunnyvale, CA, USA. AβpE3-42 was purchased from Bachem AG Bubendorf, Switzerland. 

3NTyr10Aβ was provided to us by Clinical Neuroscience Unit, Department of Neurology, University of Bonn, Sigmund-Freud-Strasse 25, 53127 Bonn, Germany. 3NTyr10Aβ was made as described in [4]. 

We included this information in lines 133, 145 – 148 (unmarked copy)/lines 154, 168 – 170 (marked-up copy) of the revised manuscript. 

o Concentration of hexafluoroisopropanol in the Abeta 1-42 solution should be provided.

For aliquotation Abeta 1-42 was dissolved in 100% hexafluoroisopropanol at a concentration of 1 mg/ml. As described above HFIP was completely removed after aliquotation and therefore no HFIP was present in the Abeta 1-42 solution that was injected. We once again apologize for the incorrect information about HFIP in the first version of the manuscript.

o What was the oligomerization status of all 4 Abeta species at the time of injection? Was it comparable?

All 4 Abeta species were injected 15 to 45 minutes after preparation. We assume that at this early timepoint after dissolving the substances the oligomerization process has just started and we injected predominantly Abeta monomers. Further oligomerization then should have taken place in the brains of the animals. A part of our study was to investigate whether this process leads to formation of Abeta plaques. As we were not able to detect Abeta plaques, retrospectively it would have been interesting to investigate for formation of Abeta oligomers e.g. using SDS-PAGE. We will consider this interesting approach when planning our next experiments.

o Concentrations of antibodies used for WB should be provided.

We apologize for not including this in the manuscript in the first place. The antibodies for WB were as follows: 

Primary antibodies 

Anti-Metabotropic Glutamate Receptor 5 Antibody ab53090 1:1000 Abcam (Cambridge, UK)

TNFα Antibody ProSci XP-5284 1:1000

 ProSci (Poway, CA, USA)

NMDAR2B Rabbit Antibody #4207 1:1000 Cell Signaling Technology (Danvers, Massachusetts, USA)

Caspase-3 Rabbit Antibody #9662 1:1000 Cell Signaling Technology (Danvers, Massachusetts, USA)

Secondary antibodies 

Anti-rabbit IgG, HRP-linked Antibody #7076 1:10 000 Cell Signaling Technology (Danvers, Massachusetts, USA)

We corrected for this in lines 227, 228, 230 (unmarked copy)/lines 249, 250, 252 (marked-up copy) of the revised manuscript. 

o Why were the hippocampus and sensory cortex chosen for WB analysis?

We wanted to investigate cognitive and behavioural changes in the modified hole-board test in wild type mice after intracerebroventricular injection of different Abeta subspecies as the primary endpoint of our study. As the hippocampus is important for working and reference memory and the sensory cortex plays a pivotal role in spatial orientation and movement planning, we decided to analyze TNF alpha, caspase 3, NR2B and mGlu5 as secondary endpoints in those brain regions.

o How many saggital slices were analysed per mouse? Which brain regions were analysed?

We analysed a total of 21 sagittal slices per mouse. The frontal and temporal lobe including sensory cortex and hippocampus were analysed. 

We included the number of brain slices and the location in line 198 (unmarked copy)/line 220 (marked-up copy) of the revised manuscript. 

Results and Interpretation

• The results are currently unclear and could be improved:

o Are the significant differences in the time trial data for the Abeta-pyro condition only for the 7-8th day of testing? Is the statistically significant difference between Abeta-pyro vs PBS for the sham or isoflurane condition or both?

Thank you very much for these important remarks. We analysed the data derived from the modified hole-board test using general linear models for the factors subspecies for injection, anaesthesia (isoflurane or sham) and the within-group factor time and their interaction terms. The results show differences that are present over the whole test period of 8 days and not only on the 7-8th day of testing. We agree with the reviewer, that the symbols marking significance in figures 1 und 2 could have been confusing for the reader, changed the figures accordingly and added the description “Effect over all days of testing” in the legend. Statistical analysis revealed a significant difference for the factor “subspecies for injection” Therefore, the difference between A-beta pyro and PBS is present for all animals. To make this clearer for the reader, we added mean difference and partial eta-squared for assessment of the effect size in the manuscript (unmarked version of the revised manuscript: lines 262, 263, 265, 275 – 278, 281 - 283/ marked-up version of the revised manuscript: lines 285, 286, 288, 289, 298 – 300, 303 - 306). 

o Are the data described in line 232 and 233 (Fig 2A) of the manuscript for the sham condition, the anaesthetic condition or both? Is the difference only present for the 7-8th day of testing?

Regarding this comment please be referred to our answer on your comment above. The difference is present over all days of testing. We corrected figure 2 accordingly.

• Abeta-pyro is by far the most concentrated Abeta species when injected, could this explain why the other Abeta species had no/limited effects? And explain the interaction of only Abeta-pyro and isoflurane in the behavioural testing data? Opposite effects of picomolar and micromolar Abeta1-42 and Abeta1-40 have previously been reported (Lazarevic et al., 2017, doi: 10.3389/fnmol.2017.00221). There are also many published papers reporting protective effects of Abeta that should be included in the discussion related to the Abeta-pyro results (i.e. Carrillo-Mora et al., 2014, doi: 10.1155/2014/795375).

Although we cannot rule out that the different concentrations of the Abeta species, with Abeta-pyro being the most concentrated could be the reason for our results, we think that the distinctive properties of Abeta pyro might explain our findings and the interaction of only Abeta-pyro and isoflurane. Please be also referred to our answer to the following critique as well as the third Reviewer comment in the Methods section. 

In order to improve the discussion of the Abeta effects we included the referenced studies in lines 351 – 354 and 371 – 374 (unmarked version)/lines 374 - 377 and 395 – 398 (marked-up version) of the revised manuscript. 

• Is there a potential mechanism for the interaction between Abeta-pyro and isoflurane to explain the results?

The fact that most of the statistically significant results in our study could be found regarding Abeta-pyro might be due to the rather short time between the intracerebroventricular injection and the behavioural testing. Abeta-pyro has been shown to accumulate in the brain at early stages of AD [2, 19], with the hippocampus being one of the predominant regions. 12-week-old C57BL/6 mice showed an impairment in spatial working memory and delayed learning in Y-maze and Morris water maze tests after intracerebroventricular injection of aggregated Abeta-pyro within two weeks after injection [20]. Abeta-pyro and isoflurane are both hydrophobic agents [21, 22]. As aggregated Abeta-pyro forms membrane pores and thus seems to alter membrane permeability [23], we also see a potential mechanism for the interaction there. 

We included our thoughts on a potential mechanism of interaction in the discussion (lines 363 – 365 (unmarked version)/lines 387 – 389 (marked-up version) of the revised manuscript). 

• Abeta1-42 is injected containing hexafluoroisopropanol and NaOH, there is no control group that accounts for the hexafluoroisopropanol and NaOH, thus, the difference in the time spent on the exposed part of the arena in the Abeta1-42 experimental group (Figure 2A) may be due to the impact of hexafluoroisopropanol, NaOH, Abeta1-42 or a combination of the three.

Once again, we are deeply sorry for not providing sufficient detail in the Methods section and would like to refer to our answer to the second reviewer comment on the methods. The Abeta 1-42 stock solution was purged from hexafluoroisopropanol (HFIP). NaOH in a concentration of 20 mmol/l was diluted 1:100 with PBS and reacted to sodium carbonate and sodium bicarbonate, so we think that the results are due to the impact of Abeta 1-42 rather than HFIP or NaOH. 

• The lack of difference in the amyloid deposits and TNFalpha, caspase 3, NR2B and mGlu5 may be due to the study being underpowered or a non-optimal concentration of Abeta species being used. This should be discussed.

We fully agree with Reviewer #1 and would also like to refer to our answer to the first and third Reviewer comment in the Methods section. The primary endpoint of the study was the cognitive and behavioural outcome. The explorative studies we did on a limited number of brains on amyloidopathy, neuroinflammation and receptor expression should be considered preliminary. We stated this in lines 32, 250, 317, 382, 411, 412 (unmarked version)/lines 34, 273, 341, 406, 436, 437 (marked-up version) of the revised manuscript. 

• What were the differences between the previous study Schmid et al., 2017 (Reference 8) and the current study?

The anaesthetic used for canula insertion surgery is different, the concentration of Abeta 1-42 injected is much higher, the number of mice per experimental group is higher, this warrants discussion.

In the previous study of Schmid et al., 2017 mice were intracerebroventricularly injected with Abeta 1-42 or PBS. Subsequently neurocognitive and behavioural parameters were evaluated using the modified hole-board test. Mice were anaesthetized with a combination of midazolam, medetomidine and fentanyl intraperitoneally. The main differences between the study under review and the previous one are the different subspecies of amyloid-beta and the type of anaesthesia: since we wanted to investigate the effects of different amyloid-beta subspecies on cognition and behaviour, mice were intracerebroventricularly injected with not only Abeta 1-42 or PBS but with Abeta 1-40, Abeta 1-42, Abeta nitro, Abeta pyro and PBS. Also, we aimed to investigate a possible interaction between isoflurane anaesthesia and the respective amyloid-beta subspecies. We decided to also use isoflurane anaesthesia for cannula implantation to avoid any interaction of other anaesthetics and opioids like midazolam, medetomidine and fentanyl with the Abeta subforms.

We agree that the concentration of Abeta 1-42 used in our 2017 study was higher. However, the final concentration of Abeta 1-42 in the brain is comparable as the amount of Abeta 1-42 is identical: In our 2017 study we injected 3.5 µl of a solution containing 1 µmol/l Abeta 1-42 resulting in a total amount of Abeta 1-42 of 3.5 pmol [24]. In the current study we injected 5.0 µl of a solution containing 700 nmol/l Abeta 1-42 resulting in the same total amount of Abeta 1-42 of 3.5 pmol. We agree that it was not clear for the reader at first sight that we injected 5.0 µl of Abeta and added the exact volume in the methods section (line 144 (unmarked version)/line 166 (marked-up copy) of the revised manuscript).

We apologize for the confusion caused regarding the experimental groups. In the study of Schmid et al, 2017, a total of 24 mice was divided in 4 groups. The first group was injected with Abeta 1-42 and started testing on day 2, the second group (also injected with Abeta 1-42) started on day 4, the third group (also injected with Abeta 1-42) started on day 8 after the intracerebroventricular injection. The fourth group (mice injected with PBS) started on day 4 after the injection. In the current study we used 60 mice divided in 10 groups. In each study we used 6 mice per group. To make this clearer for the reader we included the exact number of animals per group in the manuscript (lines 122 - 125 (unmarked copy)/lines 143 – 146 (marked-up copy)).

Other

• The manuscript should use inclusive language for people living with dementia (i.e. “patients suffering AD” is not appropriate, “people living with AD” is appropriate). This should be revised throughout. 

Thank you for this important comment. We revised the wording in lines 50, 60, 62 (unmarked copy)/lines 53, 64, 67 (marked-up copy) of the manuscript. 

• Figure 1 – reduce the range of the y axis so that data and error bars can be seen clearly.

We apologize and revised figure 1 accordingly. 

• Figure 1, 2 – marking of significance on graphs is unclear, which time point is this for? Please rectify.

We revised figures 1 and 2. Please be referred to our answer on your first comment in results and interpretation. 

• Line 232 remove the word “as”. 

We removed the word “as” in line 274 (unmarked version)/line 297 (marked-up copy) of the revised manuscript. 

• Line 268 “…an inhibiting effect on behaviour” this should be reworded to be more specific.

We apologize for not expressing this more precisely and reworded the sentence in lines 313, 314 (unmarked copy)/lines 336, 337 (marked-up copy) of the revised manuscript. 

• Paragraphs 2 and 3 of the Discussion could be integrated for a better justification of the mouse model used.

We integrated paragraphs 2 and 3 of the Discussion in lines 69 – 75 (unmarked version)/lines 70 – 85 (marked-up version) of the revised manuscript. 

• “in vitro” and “in vivo” should be italicised throughout. 

We italicised “in vitro” as well as “in vivo” in lines 80, 81, 153, 332, 347, 362, 363 (unmarked version)/in lines 94, 95, 176, 355, 370, 386, 387 (marked-up version) of the revised manuscript. 

Reviewer #2: The present study investigated the effect of different amyloid beta subspecies on behaviour and cognition in mice and their interaction with isoflurane anesthesia. The main findings of the study are that Aβ pyro improved overall cognitive performance which seemed to be contrary to the previous studies, and isoflurane could counteract this improvement. Inflammation and apoptosis biomarkers such as tumor -necrosis factor alpha, NR2B, mGlu5, or caspase 3 were not involved in this process. There are several points which the authors should consider.

1. The biochemical endpoints appear to be limited to a single evaluation This would be informative to have additional time points to better understand the effects on these markers.

We absolutely agree with Reviewer #2 that it would have been very interesting to perform biochemical analyses at several timepoints throughout our experiments. For example, given the fact that the caspase activation may not last very long following isoflurane anesthesia [7], it would have been informative to test for caspase 3 activity early after isoflurane anesthesia. However, the primary endpoint of our study was cognitive performance and behavioural alterations in the modified hole-board test. Like any other test in animals, the modified hole-board test is also very vulnerable regarding any interference. Exposing the animals to additional stressful situations, i.e. taking multiple blood samples, during the eight consecutive days of the modified hole-board testing would have altered the test performance of the mice. Therefore, we decided to evaluate the biochemical endpoints one time at the end of the mHBT being aware of this limitation to the study. To state this more clearly for the reader we reworded the corresponding paragraph in the discussion and emphasized the limitation of a missing consecutive analysis of biochemical parameters (lines 396 – 399 (unmarked version of the revised manuscript)/ lines 421 – 424 (marked-up copy of the revised manuscript)).

2. Isoflurane is rarely used clinically, as it is known for its adverse effects on cognitive functions. It would be more pellucid if the authors could illuminate the reason why selected isoflurane in this study.

Thank you for this important comment. We are sorry for not explaining this more clearly beforehand. As Reviewer #2 correctly stated, we wanted to investigate the effects of different amyloid beta subspecies on behaviour and cognition after intracerebroventricular injection in male C57BL/6N mice and their interaction with anaesthesia. We are aware, that isoflurane has adverse effects on cognitive function and is not regularly used in first world countries anymore. However, isoflurane is one of the most extensively studied anaesthetic agents in animal research. It has been shown to lead to increased oligomerization of amyloid beta in vitro [25, 26] but not in vivo [27]. Since the above cited studies used human cell lines transfected with APP or a transgenic mouse model of Alzheimer’s disease, we decided to investigate the direct interaction of isoflurane and “extrinsic” amyloid beta in a mouse model of intracerebroventricular injection. We included our considerations regarding the use of isoflurane in the introduction of the revised manuscript (lines 78 - 82 (unmarked version)/ lines 92 – 96 (marked-up version)).

We agree with the reviewer, that it would be very useful to examine different anaesthetics in future studies. Especially desflurane could be an interesting anaesthetic as it did not increase amyloid beta and tau levels in human cerebrospinal fluid and therefore could be regarded as less “neurotoxic” than isoflurane [9]. We included this consideration for future research in the manuscript (lines 404 – 406 (unmarked copy)/lines 429 – 431 (marked-up copy)).

3. The size of the experimental groups of mice in the study is unclear. Is the number of animals included based on power analysis? Please state the number of animals in each experimental group in Material and Methods

We are deeply sorry for not stating this more clearly. We included the number of animals in each experimental group (6 mice per group, resulting in a total of 60 mice) in Materials and Methods lines 122 – 125 (unmarked version)/lines 143 – 146 (marked-up version). 

Concerning a power analysis, please also be referred to Reviewer#1’s first comment in the Methods section. We apologize for not describing these very important considerations in the first version of our manuscript. In short, this study was designed as an observational study, with cognitive and behavioural outcome as primary endpoints. Therefore, sample size calculations were performed based on the following considerations: The variables of the hole-board test are considered relevant if two groups differ two times the given standard deviation. Based on a type I error of 0.05, a type II error of 0.20 and two-sided t-tests at the final test level of the hierarchical model 4 animals per group would have been appropriate. Our internal standard, however, suggests a minimal group size of six, which has been used accordingly. We added this information in lines 245 – 249 (unmarked version) of the revised manuscript/lines 268 - 272 (marked-up version) of the revised manuscript.

References

1. Mori H, Takio K, Ogawara M, Selkoe DJ. Mass spectrometry of purified amyloid beta protein in Alzheimer's disease. The Journal of biological chemistry. 1992;267(24):17082-6. Epub 1992/08/25.

2. Saido TC, Iwatsubo T, Mann DM, Shimada H, Ihara Y, Kawashima S. Dominant and differential deposition of distinct beta-amyloid peptide species, A beta N3(pE), in senile plaques. Neuron. 1995;14(2):457-66. Epub 1995/02/01.

3. Guix FX, Uribesalgo I, Coma M, Muñoz FJ. The physiology and pathophysiology of nitric oxide in the brain. Progress in neurobiology. 2005;76(2):126-52. Epub 2005/08/24.

4. Kummer MP, Hermes M, Delekarte A, Hammerschmidt T, Kumar S, Terwel D, et al. Nitration of tyrosine 10 critically enhances amyloid β aggregation and plaque formation. Neuron. 2011;71(5):833-44. Epub 2011/09/10.

5. Wirths O, Breyhan H, Cynis H, Schilling S, Demuth HU, Bayer TA. Intraneuronal pyroglutamate-Abeta 3-42 triggers neurodegeneration and lethal neurological deficits in a transgenic mouse model. Acta Neuropathol. 2009;118(4):487-96. Epub 2009/06/24.

6. Guivernau B, Bonet J, Valls-Comamala V, Bosch-Morató M, Godoy JA, Inestrosa NC, et al. Amyloid-β Peptide Nitrotyrosination Stabilizes Oligomers and Enhances NMDAR-Mediated Toxicity. The Journal of neuroscience : the official journal of the Society for Neuroscience. 2016;36(46):11693-703. Epub 2016/11/18.

7. Xie Z, Culley DJ, Dong Y, Zhang G, Zhang B, Moir RD, et al. The common inhalation anesthetic isoflurane induces caspase activation and increases Aβ level in vivo. Annals of neurology. 2008;64(6):618-27.

8. Jiang J, Jiang H. Effect of the inhaled anesthetics isoflurane, sevoflurane and desflurane on the neuropathogenesis of Alzheimer's disease (review). Mol Med Rep. 2015;12(1):3-12. Epub 2015/03/05.

9. Zhang B, Tian M, Zheng H, Zhen Y, Yue Y, Li T, et al. Effects of anesthetic isoflurane and desflurane on human cerebrospinal fluid Aβ and τ level. Anesthesiology. 2013;119(1):52-60. Epub 2013/02/27.

10. Quiroga C, Chaparro RE, Karlnoski R, Erasso D, Gordon M, Morgan D, et al. Effects of repetitive exposure to anesthetics and analgesics in the Tg2576 mouse Alzheimer's model. Neurotoxicity research. 2014;26(4):414-21. Epub 2014/06/15.

11. Kuehn BM. Anesthesia-Alzheimer disease link probed. Jama. 2007;297(16):1760. Epub 2007/04/26.

12. Culley DJ, Xie Z, Crosby G. General anesthetic-induced neurotoxicity: an emerging problem for the young and old? Current opinion in anaesthesiology. 2007;20(5):408-13. Epub 2007/09/18.

13. Jiang J, Dong Y, Huang W, Bao M. General anesthesia exposure and risk of dementia: a meta-analysis of epidemiological studies. Oncotarget. 2017;8(35):59628-37. Epub 2017/09/25.

14. Sprung J, Warner DO, Knopman DS, Petersen RC, Mielke MM, Jack CR, Jr., et al. Exposure to surgery with general anaesthesia during adult life is not associated with increased brain amyloid deposition in older adults. British journal of anaesthesia. 2020;124(5):594-602. Epub 2020/03/17.

15. Lee JJ, Choi GJ, Kang H, Baek CW, Jung YH, Shin HY, et al. Relationship between Surgery under General Anesthesia and the Development of Dementia: A Systematic Review and Meta-Analysis. BioMed research international. 2020;2020:3234013. Epub 2020/04/28.

16. Capone R, Quiroz FG, Prangkio P, Saluja I, Sauer AM, Bautista MR, et al. Amyloid-beta-induced ion flux in artificial lipid bilayers and neuronal cells: resolving a controversy. Neurotoxicity research. 2009;16(1):1-13. Epub 2009/06/16.

17. Ennaceur SM, Sanderson JM. Micellar aggregates formed following the addition of hexafluoroisopropanol to phospholipid membranes. Langmuir : the ACS journal of surfaces and colloids. 2005;21(2):552-61. Epub 2005/01/12.

18. Rammes G, Seeser F, Mattusch K, Zhu K, Haas L, Kummer M, et al. The NMDA receptor antagonist Radiprodil reverses the synaptotoxic effects of different amyloid-beta (Aβ) species on long-term potentiation (LTP). Neuropharmacology. 2018;140:184-92. Epub 2018/07/18.

19. Sergeant N, Bombois S, Ghestem A, Drobecq H, Kostanjevecki V, Missiaen C, et al. Truncated beta-amyloid peptide species in pre-clinical Alzheimer's disease as new targets for the vaccination approach. Journal of neurochemistry. 2003;85(6):1581-91. Epub 2003/06/06.

20. Youssef I, Florent-Béchard S, Malaplate-Armand C, Koziel V, Bihain B, Olivier JL, et al. N-truncated amyloid-beta oligomers induce learning impairment and neuronal apoptosis. Neurobiol Aging. 2008;29(9):1319-33. Epub 2007/04/27.

21. Perez-Garmendia R, Gevorkian G. Pyroglutamate-Modified Amyloid Beta Peptides: Emerging Targets for Alzheimer´s Disease Immunotherapy. Current neuropharmacology. 2013;11(5):491-8. Epub 2014/01/10.

22. Pavel MA, Petersen EN, Wang H, Lerner RA, Hansen SB. Studies on the mechanism of general anesthesia. Proceedings of the National Academy of Sciences of the United States of America. 2020;117(24):13757-66. Epub 2020/05/30.

23. Piccini A, Russo C, Gliozzi A, Relini A, Vitali A, Borghi R, et al. beta-amyloid is different in normal aging and in Alzheimer disease. The Journal of biological chemistry. 2005;280(40):34186-92. Epub 2005/08/17.

24. Schmid S, Jungwirth B, Gehlert V, Blobner M, Schneider G, Kratzer S, et al. Intracerebroventricular injection of beta-amyloid in mice is associated with long-term cognitive impairment in the modified hole-board test. Behav Brain Res. 2017;324:15-20. Epub 2017/02/15.

25. Xie Z, Dong Y, Maeda U, Alfille P, Culley DJ, Crosby G, et al. The common inhalation anesthetic isoflurane induces apoptosis and increases amyloid beta protein levels. Anesthesiology. 2006;104(5):988-94. Epub 2006/04/29.

26. Eckenhoff RG, Johansson JS, Wei H, Carnini A, Kang B, Wei W, et al. Inhaled anesthetic enhancement of amyloid-beta oligomerization and cytotoxicity. Anesthesiology. 2004;101(3):703-9. Epub 2004/08/27.

27. Bianchi SL, Tran T, Liu C, Lin S, Li Y, Keller JM, et al. Brain and behavior changes in 12-month-old Tg2576 and nontransgenic mice exposed to anesthetics. Neurobiology of aging. 2008;29(7):1002-10.

---

## [Editor Report · Decision Letter 1]

13 Nov 2020

Neurotoxicity of different amyloid beta subspecies in mice and their interaction with isoflurane anaesthesia

PONE-D-20-10009R1

Dear Dr. Schmid,

We’re pleased to inform you that your manuscript has been judged scientifically suitable for publication and will be formally accepted for publication once it meets all outstanding technical requirements.

Kind regards,

Stephen D. Ginsberg, Ph.D.

Section Editor

PLOS ONE

---

## [Editor Report · Acceptance letter]

25 Nov 2020

PONE-D-20-10009R1 

Neurotoxicity of different amyloid beta subspecies in mice and their interaction with isoflurane anaesthesia 

Dear Dr. Schmid:

I'm pleased to inform you that your manuscript has been deemed suitable for publication in PLOS ONE. Congratulations! Your manuscript is now with our production department. 

Kind regards, 

on behalf of

Dr. Stephen D. Ginsberg 

Section Editor

PLOS ONE